# Context-Agnostic Learning Using Synthetic Data

## Abstract

We propose a novel setting for learning, where the input domain is the image of a map defined on the product of two sets, one of which completely determines the labels. Given the ability to sample from each set independently, we present an algorithm that learns a classifier over the input domain more efficiently than sampling from the input domain directly. We apply this setting to visual classification tasks, where our approach enables us to train classifiers on datasets that consist entirely of a single example of each class. On several standard benchmarks for real-world image classification, our approach achieves performance competitive with state-of-the-art results from the few-shot learning and domain transfer literature, while using significantly less data.

## 1 Introduction

Despite recent advances in deep learning, one central challenge is the large amount of labelled training data required to achieve state-of-the-art performance. Procuring such volumes of high quality, reliably annotated data can be costly or even close to impossible (e.g., obtaining data to train an autonomous navigation system for a lunar probe). Additional hurdles include hidden biases in large datasets (Tommasi et al., 2017) and maliciously perturbed training data (Biggio et al., 2012).

Synthetically generated data has seen growing adoption in response to these problems, since the marginal cost of producing new training data is generally very low, and one has full control over the generation process. This is particularly true for applications with a physical component, such as autonomous navigation (Gaidon et al., 2016) or robotics (Todorov et al., 2012). However, training with purely synthetic data suffers from the so-called "reality gap", whereby good performance on synthetic data does not necessarily yield good performance in the real world (Jakobi et al., 1995). In particular, the difficulty of generating realistic training images scales not just with the objects of interest, but also the real-world contexts in which the learned model is expected to operate.

This work begins with the simple observation that, for many classification tasks, the label of an input is determined entirely by the object; however, this additional structure is discarded by current synthetic data pipelines. Our goal is to leverage this decomposition to develop more efficient methods for the related problems of generating training data and learning from a synthetic domain.

Our contributions are two-fold: first, we formally introduce the setting of context-agnostic learning, where the input space is decomposed into object and context spaces, and the labels are independent of contexts when conditioned on the objects. Second, we propose an algorithm to efficiently train a classifier in the context-agnostic setting, which relies on the ability to sample from the object and context spaces independently. We apply our methods to train deep neural networks for real-world image classification using only a single synthetic example of each class, obtaining performance comparable to existing methods for domain adaptation and few-shot learning while using substantially less data. Our results show that it is possible to train classifiers in the absence of any contextual training data that nonetheless generalize to real world domains.

## 2 Related work

Domain shift refers to the problem that occurs when the training set (source domain) and test set (target domain) are drawn from different distributions. In this setting, a classifier which performs

well on the source domain may not generalize well in the target domain. A standard method for addressing this challenge is domain adaptation, which leverages a small amount of data from the target domain to adapt a function that is learned over the source domain (Blitzer et al., 2006).

In the context of learning from synthetic data, the domain shift that occurs between synthetic and real world data is known as the reality gap (Jakobi et al., 1995). State-of-the-art rendering engines, such as those used for video games, can help narrow this gap by generating photorealistic data for training (Dosovitskiy et al., 2017; Johnson-Roberson et al., 2016; Qiu and Yuille, 2016). Another technique is using domain randomization to generate the source domain with more variability than is expected in the target domain (e.g., extreme lighting conditions and camera angles), so as to make real images appear as just another variant (Tobin et al., 2017; Tremblay et al., 2018); in particular, Torres et al. (2019) apply domain randomization to traffic sign detection and find that arbitrary natural images suffice for the task. Another body of work exploits generative adversarial networks (Goodfellow et al., 2014a) to generate synthetic domains (Hoffman et al., 2017; Liu et al., 2017; Shrivastava et al., 2016; Taigman et al., 2016; Tzeng et al., 2017). Finally, several works have explored using synthetic data for natural image text recognition (Gupta et al., 2016; Jaderberg et al., 2014). These works use an approach that is roughly analogous to our baseline models, and test their techniques on the target domain of street signs rather than handwritten characters (as we do).

A different paradigm for the low-data regime is few-shot learning. In contrast to domain adaptation, few-shot learning operates under the assumption that the target and source distributions are the same, but the ability to sample certain classes is limited in the source domain. Early approaches emphasized capturing knowledge in a Bayesian framework (Fe-Fei et al., 2003), which was later formulated as Bayesian program learning (Lake et al., 2015). Another approach based on metric learning is to find a nonlinear embedding for objects where closeness in the geometry of the embedding generalizes to unseen classes (Koch, 2015; Snell et al., 2017; Sung et al., 2018; Vinyals et al., 2016). Meta-learning approaches aim to extract higher level concepts which can be applied to learn new classes from a few examples (Finn et al., 2017; Munkhdalai and Yu, 2017; Nichol et al., 2018; Ravi and Larochelle, 2016). A conceptually-related method that leverages synthetic training data is learning how to generate new data from a few examples of unseen classes; in contrast to our work, however, these methods still require a large number of samples to learn the synthesizer (Schwartz et al., 2018; Zhang et al., 2019). Finally, some works combine domain adaptation with few-shot learning to learn under domain shift and limited samples (Motiian et al. (2017)).

The main characteristic that differentiates our work from these approaches is that we are interested in learning classifiers that are *context-agnostic*, i.e., do not rely on background signals. As such, while we find our approach is applicable to many of the same tasks as the aforementioned works, our theoretical setting and objectives differ significantly. From a practical perspective, we demonstrate our techniques when *the entire training set consists solely of a single synthetic image of each class*, though our techniques can certainly be applied when more data is available; however we do not expect the reverse to hold for domain adaptation or few-shot learning in our setting. Indeed, we consider this work to be complementary in that we are concerned with exploiting the additional structure that is inherent in certain source domains, while the goal of domain adaptation and few-shot learning is to achieve good performance under various downstream domain shift assumptions.

## 3 SETTING

The standard supervised learning setting consists of an input space $\mathcal{X}$, an output space $\mathcal{Y}$, and a hypothesis space $\mathcal{H}$ of functions mapping $\mathcal{X}$ to $\mathcal{Y}$. A domain $P_D$ is a probability distribution over $(\mathcal{X}, \mathcal{Y})$. Given a target domain $P_T$ and a loss function $\ell$, the goal is to learn a classifier $h \in \mathcal{H}$ that minimizes the risk, i.e., the expected loss $R_{P_T}(h) := \mathbb{E}_{P_T}[\ell(h(x), y)]$. The training procedure consists of $n$ samples $(x_1, y_1), ..., (x_n, y_n)$ from a source domain $P_S$. A standard approach is empirical risk minimization, which takes the classifier that minimizes $R_{emp}(h) = \frac{1}{n} \sum_i \ell(h(x_i), y_i)$; if $P_S$ is close to $P_T$, then with enough samples, such a classifier also achieves low risk in the target domain.

### 3.1 CONTEXT-AGNOSTIC LEARNING

In general, we can frame the goal of classification as learning to extract reliable signals for the label $y$ from points $x \in \mathcal{X}$. This task is often complicated by the presence of noise or other spurious signals.

However, for input spaces generated by physical processes, such signals are generally produced by distinct physical entities and can thus be thought of as independent signals that become mixed via the observation process. We aim to capture this additional structure in our setting.

Concretely, we have an object space $\mathcal{O}$, a context space $\mathcal{C}$, and an observation function $\gamma$ on $\mathcal{O} \times \mathcal{C}$. The input space $\mathcal{X}$ is defined as the image of $\gamma : \mathcal{O} \times \mathcal{C} \to \mathcal{X}$. We will assume that points in $\mathcal{O}$ are associated with a unique label in $\mathcal{Y}$, and require that $\gamma$ preserves this property when passing to $\mathcal{X}$. Note that this setting can be easily generalized to a case when the image of $\gamma$ is a subdomain of $\mathcal{X}$.

In this work, we will consider the special case when $\mathcal{X} \subseteq \mathcal{C}$. Conceptually, the context space is an "ambient space" containing not only valid inputs, but also random noise or irrelevant classes; the input space is a subset of the context space for which there exists a well-defined label. For example, in our experiments we explore such a decomposition for the task of traffic sign recognition, where the object space $\mathcal{O}$ consists of traffic signs viewed from different angles, the context space $\mathcal{C}$ is unconstrained pixel space, and the input space $\mathcal{X}$ is the set of images that contain a traffic sign.

Recall that the standard objective of learning is to find a good classifier for an unknown subdomain $\mathcal{X}_{P_T} \subseteq \mathcal{X}$. We consider instead the task of learning a classifier on the entire input space $\mathcal{X}$. To sample from $\mathcal{X}$ we are given oracle access to the observation function and draw (labelled) samples from $\mathcal{O}$ and $\mathcal{C}$ independently. Clearly, if this problem is realizable, i.e., there exists $h^* \in \mathcal{H}$ for which $R_{\mathcal{X}}(h^*) = 0$, then we do not even need to know the target domain $P_T$, since

$$\mathcal{X}_{P_T} \subseteq \mathcal{X} \implies \left[ R_{\mathcal{X}}(h^*) = 0 \implies R_{\mathcal{P}_T}(h^*) = 0 \right]$$

Assuming access to $\mathcal{X}$ through $\gamma$, we can learn $h^*$ simply by taking the number of samples to infinity. Unfortunately, learning a classifier on $\mathcal{X}$ generally requires many more samples than learning a classifier on $\mathcal{X}_{P_T}$. Thus we aim to learn $h^*$ using as few samples as possible.

Our new goal will be to learn a classifier over $\mathcal{X}$ which depends only on signals from $\mathcal{O}$; more precisely, we have the following definitions:

**Definition 3.1.** *A function $f$ on $\mathcal{X}$ is **context-agnostic** if*

$$\Pr[f \circ \gamma(o, c) = x] = \Pr[f \circ \gamma(o, c') = x] \qquad \forall c, c' \in \mathcal{C}, o \in \mathcal{O}, x \in \mathrm{Im}(f)$$

**Definition 3.2.** *Given a context-agnostic label function $y^*$, the objective of **context-agnostic learning** is to find $h \in \mathcal{H}$ such that $h$ achieves the lowest risk of all context-agnostic classifiers.*

The hope is that, since $y^*$ is context-agnostic, we can learn $y^*$ through the lower dimensional structure of $\mathcal{O}$ using fewer samples. Note, however, that while we only need $\max(|\mathcal{O}|, |\mathcal{C}|)$ samples to observe every object and context once, we need $|\mathcal{O}| * |\mathcal{C}|$ samples to observe every object in every context. Hence the main challenge when the number of samples is low will be avoiding *spurious signals*, i.e., statistical correlations between context and objects (and by extension, labels) which are artifacts of the sampling process and do not generalize outside the training set.

We conclude with some high-level remarks about this setting. First, note that if the problem is realizeable, then the lowest risk classifier is also context-agnostic. Second, we recover the standard supervised setting for the trivial context space $\mathcal{C} = \emptyset$. Conversely, classification remains well-defined even in the trivial object space $\mathcal{O} = \{y_i\}$, the set of classes; however, this pushes all the complexity to the observation function $\gamma$, which may be hard to define or intractable to compute. Finally, we do not preclude the existence of useful signals originating from the context for certain domains. For instance, a great deal of information can often be gleaned from the backgrounds of photos, e.g., stop signs are more often found in cities than on highways. Our theoretical setting avoids this issue by assuming realizability and uniqueness of labels; more practically, we argue that a "good" classifier should nonetheless recognize stop signs on the highway, and our experimental results provide evidence that over-reliance on such background signals leads to brittle classifiers.

## 3.2 EFFICIENT SAMPLING FOR OBJECT-CONTEXT DECOMPOSED INPUT SPACES

In this section, we present an algorithm for context-agnostic learning. We first develop a formal notion of contextual bias for this setting. We assume a binary classifier $h$ and slightly abuse notation, writing $h$ for $h \circ \gamma$, i.e., $h : \mathcal{O} \times \mathcal{C} \to \{-1, 1\}$. For an object $o$, denote the correct label $o^*$, the expected classification $\bar{o} := \mathbb{E}_{c \sim \mathcal{C}}[h(o, c)]$, and the object error $\hat{o} := |o^* - \bar{o}|$.

**Definition 3.3.** *The **context bias** $B(h, c)$ of a classifier $h$ on the context $c$ is defined as*

$$sgn(B(h, c)) := sgn(\mathbb{E}_{o \sim \mathcal{O}}[h(o, c) - \bar{o}])$$
$$||B(h, c)|| := \mathbb{E}_{o \sim \mathcal{O}}\big[\ell\big(h(o, c), \bar{o}\big)\big]$$

*where $\ell$ is the hinge loss $\ell(i, j) := \max(0, 1 - i * j)$.*

Intuitively, sign of the bias corresponds to the label toward which the classifier is biased by a given context; the magnitude measures the strength of this bias. Clearly, the classifier is context-agnostic exactly when the bias is zero. We are now ready to state our main theoretical result, which gives an upper bound on the risk in terms of the context bias on $\mathcal{C}$ and object error over $\mathcal{O}$.

**Theorem 3.1.** *Let $h$ be a classifier with average bias $K$ and object error for all objects bounded from above by $\alpha < 1$. Then the risk is bounded from above by $K/(2 - \alpha)$. Furthermore, equality holds if and only if all object errors equal $\alpha$.*

We give a proof in Appendix A. The assumption $\alpha < 1$ is fairly weak, being equivalent to the classifier performing better than random guessing. Note that the error bound $\alpha$ and bias bound $K$ are not independent; in particular, $\alpha = 0$ if and only if $K = 0$ and $\alpha < 1$. Observe also that when $\mathcal{C} = \emptyset$, $K = 0$ holds trivially, but $\alpha < 1$ for all objects means the classifier is correct on all inputs.

The central idea behind Theorem 3.1 is leveraging the fact that labels depend only on objects to factor the risk into separate terms for object error and context bias. This factorization enables us to exploit our ability to sample independently from the object and context spaces. More specifically, we can use samples from $\mathcal{O}$ to minimize the object error, and samples from $\mathcal{C}$ to minimize the context bias. Since we only need $\alpha < 1$, we continue to draw objects randomly; however given an object $o$, we aim to observe it with the context for which the classifier has the strongest opposing bias. Intuitively, this allows the classifier to "correct" its bias and unlearn the spurious signals, thereby minimizing the bias and also the risk.

Adopting this approach without modification requires computing the bias of every context in $\mathcal{C}$. In most cases, however, even estimating a single bias may be prohibitively expensive. Thus, rather than solve for the maximum bias explicitly, we instead propose a heuristic for identifying contexts with large biases. Note that since $\mathcal{X} \subseteq \mathcal{C}$, a reasonable assumption is that the classifier learns a strong bias on recent training inputs when taken as contexts. This suggests a simple greedy approach for correcting biases by repurposing recent training inputs as contexts; we call this algorithm Greedy Bias Correction and present a description in Algorithm 1.

## 4   LEARNING VISUAL TASKS USING CONTEXT-AGNOSTIC SYNTHETIC DATA

We introduce an instantiation of Greedy Bias Correction for learning visual tasks using synthetic data. We are given a function which takes a label $y$ and outputs a rendering of the corresponding class in a random pose without any background. The context is the background of the image, on which we place no restrictions. The observation function $\gamma$ superimposes an object over a background.

**Local refinement via robustness training**   We note that our observation function $\gamma$ is fairly restrictive; for instance, we do not support occlusions. Because our ultimate goal will be to perform on data taken from a real-world context, we aim to capture this discrepancy using robustness training.[1] In particular, we assume that the image of $\gamma$ is an $\epsilon$-covering of $\mathcal{X}$, where a set $A$ is said to be an $\epsilon$-covering of another set $B$ iff for all points $b \in B$, there exists a point $a \in A$ such that $||a - b|| \leq \epsilon$. Then for a given sample, we will instead add the point in the $\epsilon$-neighborhood of $x$ which maximizes the training loss, i.e., for a classifier $h$ and a sample $x = \gamma(o, c)$, we use $x' = \arg\max_{x' \in N_\epsilon(x)} \ell(h(x'), y)$. This formulation is often used to train models which are robust against local perturbations. An empirically effective method for finding approximations to $x'$ is known as Projected Gradient Descent (PGD) (Goodfellow et al., 2014b; Madry et al., 2017). The

---

[1]Robustness training is more commonly referred to as adversarial training in the adversarial robustness community whence we borrow this technique. We use the nonstandard term to avoid confusion with the unrelated (generative) adversarial methods found in the few-shot learning literature.

---

**Algorithm 1:** Greedy Bias Correction

**Input:** Object space $\mathcal{O}$, context space $\mathcal{C}$, observation function $\gamma$, number of rounds $R$, resample probability $p$, classifier update subroutine `Fit`, binary classifier $h$
**Output:** Trained classifier $h$

*// initialize random context and label*
$c \sim \mathcal{C}$;
$y \sim \{-1, 1\}$;
**for** $r \leftarrow 1$ **to** $R$ **do**
    $o \sim \mathcal{O}(y)$; *// sample object*
    $x \leftarrow \gamma(o, c)$; *// observe object and context*
    $h \leftarrow \texttt{Fit}(h, x, y)$; *// perform classifier update*
    *// update context and label*
    $p' \leftarrow \texttt{Uniform}(0, 1)$;
    **if** $p' < p$ **then**
        *// resample random context and label*
        $c \sim \mathcal{C}$;
        $y \sim \{-1, 1\}$;
    **else**
        $c \leftarrow x$; *// previous image becomes new context*
        $y \leftarrow -y$; *// flip label*
    **end**
**end**

---

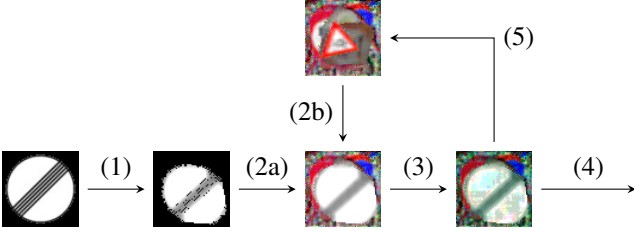

Figure 1: A graphical representation of the generative loop in Algorithm 2 using real training data. (1) Sample from object space. (2) Observe object and context. (3) Perform local refinement. (4) Add to training set. (5) Previous image becomes next context (resample from $\mathcal{C}$ with probability $p$).

algorithm can be summarized as

$$x_0 \leftarrow x + \delta$$
$$x_i \leftarrow \Pi_{x+\epsilon}\big(x_{i-1} + \eta \cdot \text{sgn}(\nabla_x \ell(h(x_{i-1}), y))\big), \quad i = 1, ..., n$$

where $\delta$ is a small amount of random noise, $\Pi$ is the projection back onto to the $\epsilon$-ball, $\eta$ is the step size, and $n$ is the number of iterations. As is standard for robustness training, we use the $\ell_\infty$ norm defined as $||(x_1, ..., x_n)||_\infty = \max_i x_i$. Our choice of $\epsilon$ will depend on the task at hand, and we also use different $\epsilon$ for the portions of the image corresponding to the object and context.

Additionally, since we are no longer in a binary context, we sample a random permutation on labels instead of flipping the label deterministically. The full algorithm is presented as Algorithm 2 in Appendix B; Figure 1 provides a visualization of the key generative process, with images taken from a real step of training a deep neural network to perform classification of traffic signs.

From a practical standpoint, this algorithm makes concrete several benefits of our approach. First, rendering object classes, i.e. sampling from $\mathcal{O}$, is often relatively easy. In the case of two-dimensional rigid body objects, this can be captured using standard data augmentation such as rotations, flips, and perspective distortions. Indeed, in this setting, our work can be viewed as a form of minimal one-shot learning, where the training data consists solely of a single unobstructed straight-on shot for each object class. Second, there is no requirement to perform realistic rendering of contexts $\mathcal{C}$, avoiding an additional layer of complexity.

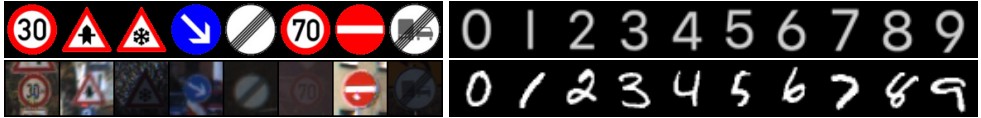

Figure 2: Images from the training (top) and test (bottom) set for GTSRB (left) and MNIST (right).

Table 1: Performance of Algorithm 2 on various benchmarks, plus ablation studies.

| Approach | Picto → GTSRB | Digit → MNIST | Omnifont → Omniglot |
|---|---|---|---|
| baseline | 72.0 | 81.9 | 71.9 |
| + random-context | 72.1 | 88.3 | 69.8 |
|   + refinement-only | 86.4 | 89.7 | 90.8 |
| + bias-correction | 87.3 | 89.2 | 80.5 |
| **+ full** | **95.9** | **90.2** | **92.2** |

Finally, because our approach is context agnostic, our functions are learned without any reference to target domains. In the formal setting, we assumed that the target domain was contained in the image of the observation function; however, synthetic images will always be subject to the reality gap. Our experiments suggest that our approach overcomes this barrier and successfully generalizes to natural images while training on synthetic data only.

## 5 EXPERIMENTS

We evaluate our approach to learning visual tasks using synthetic data on three benchmarks for image recognition. Our training sets consist of a single synthetic image for each object class with no additional information about the target domain; Figure 2 shows examples of the training and test images from two of the datasets. On all three benchmarks, our models perform comparably with previous state-of-the-art results from related settings using few-shot learning and domain adaptation. Table 1 provides a summary of our results; comprehensive results and comparisons are compiled in Appendix D. Appendix C provides the full experimental setup and training details. Sample images from all datasets referenced below, including examples of rendered training data from the experiments and ablation studies, are shown in Appendix E.

### 5.1 GTSRB

The German Traffic Sign Recognition Benchmark (**GTSRB**) (Stallkamp et al., 2012) contains 39,209 training and 12,630 test images of 43 classes of German traffic signs taken from the real world. Our training set consists of a single, canonical pictogram of each class taken from the visualization software accompanying the dataset, which we refer to as **Picto**. We achieve 95.9% accuracy on the GTSRB test set training only on Picto, against a human baseline of 98.8%. A comprehensive comparison with existing approaches can be found in Appendix D, Table 2.

**SynSign** (Moiseev et al., 2013) is a synthetic dataset designed to provide realistic training data for traffic sign recognition. The dataset comprises 100,000 synthetically generated images of signs from Sweden, Germany, and Belgium in a variety of poses, rendered against domain-appropriate real-world backgrounds (e.g. trees, roads, sky). The dataset contains a superset of the GTSRB classes; as a result, Saito et al. (2017) report 79.2% accuracy by training directly on SynSign.

For domain adaptation, all approaches train on the full 100,000 images in SynSign plus part of the GTSRB training set. ATT (Saito et al., 2017) is the only method with better performance than ours, achieving 0.3% higher accuracy; however they use 31,367 unlabelled images from the GTSRB training set (in addition to SynSign). Methods using few-shot learning train on roughly half of the data (22 classes) from the GTSRB training set. The leading few-shot learning approach, VPE (Kim et al., 2019), adds a pictographic dataset similar to Picto, but achieves only 83.79% accuracy. In comparison, our training set consists of only 43 images, none of which are from GTSRB.

## 5.2 HANDWRITTEN CHARACTER RECOGNITION

**MNIST** (LeCun) consists of 60,000 training and 10,000 test images of handwritten Arabic numerals in grayscale against a blank background. Our training set, **Digit**, consists of a single example of each digit taken from a standard digital font. **Omniglot** (Lake et al., 2015) consists of 1623 hand-written characters from 50 different alphabets, with 20 samples each. The samples were sourced online from 20 workers on Amazon's Mechanical Turk, who were asked to copy each character from a single font-based example using digital input (e.g., a mouse). We obtained the original representations for our dataset, **OmniFont**. On MNIST, we achieve 90.2% accuracy training only on Digit, compared to human accuracy of 98%; on Omniglot, we achieve 92.2% 20-way accuracy training only on Omnifont, compared to human accuracy of 95.5%. Tables 3 and 4 in Appendix D compare these results with approaches using few-shot learning and domain adaptation.

Handwritten characters and GTSRB present conceptually opposed challenges for learning: in GTSRB, the objects are rigid two-dimensional objects and backgrounds are complex settings in the natural world; in Omniglot and MNIST, backgrounds are uniform, but classes no longer have a strict specification and individual examples exhibit high variability. Thus, the main challenge of these tasks is learning how to generalize over the object class. Despite the inherent variation, a baseline model trained on Digit with plain data augmentation was able to achieve 81.9% accuracy on MNIST, exceeding many domain adaptation approaches and all the one-shot learning results; Omniglot is more difficult, with an Omnifont plus data augmentation baseline accuracy of 71.9%.

On MNIST, every approach using domain adaptation uses the full Street View House Numbers (**SVHN**) training set of 73,257 images of house numbers obtained from Google Street View (Netzer et al., 2011), plus varying amounts of data from MNIST. The domain transfer problem faces a similar challenge as Digit, namely, handwriting exhibits different characteristics than house numbers fonts. Nevertheless, we note that SVHN contains far more examples of each digit. The only non-baseline approach to exceed our performance is CyCADA (Hoffman et al., 2017), which achieves 0.2% better performance by performing domain adaptation using 60,000 unlabelled images from the MNIST training set (in addition to training on SVHN). All approaches using few-shot learning (except FADA) train on 32,460 images from Omniglot and use as few as one image per class from MNIST; the best result achieves accuracy 3% below ours using 70 images from MNIST. In contrast, we use only 10 images, none of which are from MNIST.

Omniglot is often described as an MNIST-transpose, where the goal is learn handwriting rather than specific symbols, and is widely used as a benchmark for few-shot learning. We reproduce the most common split given in Lake et al. (2015), which uses a predefined set of 30 alphabets, with 19,280 images for training. Test performance is reported as an average over random subsets of $n = 5, 20$ unseen classes for the $n$-way task (given one labelled example). In comparison, for each test run, we retrain a model using only the corresponding $n$ images from OmniFont. As expected, our method finds 5-way classification easier than 20-way classification (95.8% vs 92.2%). In both cases, our performance lags behind the state-of-the-art for few-shot learning ($>$99%), though we emphasize that our experimental setup differs significantly in both the type and amount of training data used.

Finally, several approaches apply few-shot learning from Omniglot to MNIST, with the idea of transferring extracted features from human handwriting. However the one-shot experiments all perform worse than even our baseline approach. We hypothesize that in comparison to Omniglot, where all the samples come from the same 20 subjects, MNIST may be particularly difficult for transfer one-shot learning, since any two examples will likely exhibit high "variance"; conversely, our approach benefits from using a canonical form which might be closer to the "mean" representation.

## 5.3 ABLATION STUDIES

We conduct two sets of ablation studies to better understand our approach to context-agnostic learning. The first study tests the individual components of our algorithm for their contributions to generalization over the real world dataset. All strategies employ the same data augmentation and use the following sampling procedures: **baseline** picks a fresh random background for each training point, and measures the performance of training on our synthetic dataset with plain data augmentation; **random-context** reuses random backgrounds as contexts; **bias-correction** reuses previous training images as contexts; **refinement-only** is the same as random-context with the addition of PGD-based refinement; **full** is the full algorithm as described in Algorithm 2. The results are in Table 1.

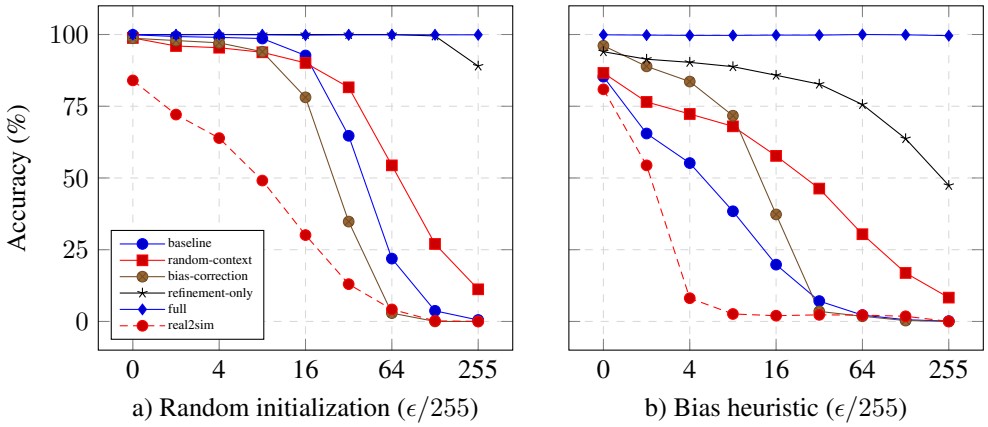

Figure 3: Context-agnostic performance on Picto using a `PGD` adversary on the background.

In all cases, we observe that both bias correction and local refinement contribute individually and jointly to the performance of our models. For GTSRB, a particularly interesting comparison is training on SynSign, a dataset designed to provide synthetic training data with realistic backgrounds for GTSRB, which yields 79.2% accuracy (Saito et al., 2017). Though this is an improvement over our baseline of using random backgrounds at 72.0% accuracy, refinement-only and bias-correction achieve higher accuracy at 86.4% and 87.3%, respectively. Both methods leverage the background of training images to combat spurious signals, generating completely unrealistic backgrounds; this suggests that learning context-agnostic features is more effective than using realistic backgrounds.

The second study measures classification performance in a context-agnostic setting on the synthetic Picto dataset. By definition, the performance of a context-agnostic classifier should not degrade under perturbations of the background. We thus run an adaptive attack using a `PGD` adversary which fixes the foreground pixels, and ranges from fixed to unbounded on the background pixels, effectively searching the context space for a background that causes a misclassification on the given object. We also consider two initialization strategies for the `PGD` adversary: a standard random initialization, and initializing to the previous image, inspired by our bias heuristic. We test the same set of strategies as before, plus a classifier trained directly on the GTSRB training set achieving 98% performance on the GTSRB test set (**real2sim**). Appendix E.2 contains samples of the generated images, and the results are plotted in Figure 3.

Across all experiments, the models have worse (or very close) performance when using our bias heuristic for initialization. We believe this supports our usage of the bias heuristic for context-agnostic learning. Additionally, in the last column of Figure 3b, only our full method maintains passable accuracy, which suggests the gap between models is larger than performance on GTSRB indicates. We also note that real2sim seems to suffer from a "synthetic gap" even at $\epsilon = 0/255$, which is not entirely unexpected. However, in both settings, performance degrades very quickly as $\epsilon$ increases: the effect is most pronounced when the bias heuristic is used to initialize the `PGD` adversary, though in both cases the accuracy eventually drops to 0. We emphasize that all of the experiments leave the foreground objects completely unperturbed (and easily human-identifiable); our results thus suggest that classifiers trained on natural images can become over-reliant on contextual signals, leading to surprisingly brittle behavior even given unambiguous foregrounds.

## 6 CONCLUSION

We introduce the task of context-agnostic learning, a theoretical setting for learning models whose predictions are independent of background signals. Leveraging the ability to sample objects and contexts independently, we propose an approach to context-agnostic learning by minimizing a formally defined notion of context bias. Our algorithm has a natural interpretation for training classifiers on vision-based tasks using synthetic data, with the distinct advantage that we do not need to model the background. We evaluate our methods on several real-world domains; our results suggest that our approach succeeds in learning context-agnostic classifiers that generalize to natural images using

only a single synthetic image of each class, while training with natural images can lead to brittleness in the context-agnostic setting. Our performance is competitive with existing methods for learning when data is limited, while using significantly less data. More broadly, the ability to learn from single synthetic examples of each class also affords fine-grained control over the data used to train our models, allowing us to sidestep issues of data provenance and integrity entirely.

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

# A    PROOFS

*Proof of Theorem 3.1.* By the assumption that $\alpha < 1$, we have that for all $o$, the signs of the expected classification $\bar{o}$ and correct classification $o^*$ match, so that $\alpha \geq \hat{o} = |o^* - \bar{o}| = 1 - |\bar{o}|$. Then for all $o$,

$$
\begin{aligned}
\ell(\bar{o}, o^*) &= 1 - \bar{o} * o^* \\
&= 1 - |\bar{o}| \\
&= \frac{1 + |\bar{o}|}{1 + |\bar{o}|}(1 - |\bar{o}|) \\
&= \frac{1 - \bar{o} * \bar{o}}{1 + |\bar{o}|} \\
&= \frac{\ell(\bar{o}, \bar{o})}{1 + |\bar{o}|} \\
&\leq \frac{\ell(\bar{o}, \bar{o})}{2 - \alpha}
\end{aligned}
$$

Now to bound the risk, we can write,

$$
\begin{aligned}
R(h) &:= \mathbb{E}_{o \sim O, c \sim C}[\ell(h(o, c), o^*)] \\
&= \frac{1}{|\mathcal{C}||\mathcal{O}|} \sum_c \sum_o \ell(h(o, c), o^*) \\
&= \frac{1}{|\mathcal{O}|} \sum_o \frac{1}{|\mathcal{C}|} \sum_c (1 - h(o, c) * o^*) \\
&= \frac{1}{|\mathcal{O}|} \sum_o (1 - \bar{o} * o^*) \\
&\leq \frac{1}{|\mathcal{O}|} \sum_o \frac{1 - \bar{o} * \bar{o}}{2 - \alpha} \\
&= \frac{1}{(2 - \alpha)|\mathcal{O}|} \sum_o \frac{1}{|\mathcal{C}|} \sum_c (1 - h(o, c) * \bar{o}) \\
&= \frac{1}{(2 - \alpha)|\mathcal{C}||\mathcal{O}|} \sum_c \sum_o \ell(h(o, c), \bar{o}) \\
&= \frac{1}{(2 - \alpha)|\mathcal{C}|} \sum_c ||B(h, c)|| \\
&= \frac{K}{2 - \alpha}
\end{aligned}
$$

as desired. It also follows that equality holds if and only if $\alpha = \hat{o}$ for all $o$. $\qquad\square$

## B   Greedy Bias Correction for visual tasks

---

**Algorithm 2:** Visual Learning Using Context-Agnostic Synthetic Data

---

**Input:** Object space $\mathcal{O}$, context space $\mathcal{C}$, random permutations $\Pi$, observation function $\gamma$, number of rounds $R$, batch size $B$, number of classes $N$, resample probability $p$, classifier update subroutine $\texttt{Fit}$, projected gradient descent subroutine $\texttt{PGD}$, classifier $h$

**Output:** Trained classifier $h$

**for** $r \leftarrow 1$ **to** $R$ **do**
    *// initialize empty training batch and random contexts*
    $X \leftarrow \emptyset$;
    **for** $n \leftarrow 1$ **to** $N$ **do**
        $c_n \sim \mathcal{C}$;
    **end**
    **for** $b \leftarrow 1$ **to** $B$ **do**
        *// sample random permutation*
        $\pi \sim \Pi(N)$;
        *// generate new training data*
        **for** $n \leftarrow 1$ **to** $N$ **do**
            $o \sim \mathcal{O}(n)$; *// sample object for class*
            $x \leftarrow \gamma(o, c_{\pi(n)})$; *// observe object and random (permuted) context*
            $x' \leftarrow \texttt{PGD}(h, x)$; *// perform local refinement*
            $X \leftarrow X \cup \{(x', y)\}$; *// add to training set*
            $c_n \leftarrow x'$; *// previous sample becomes next context*
        **end**
        *// resample contexts*
        **for** $n \leftarrow 1$ **to** $N$ **do**
            $p' \leftarrow \texttt{Uniform}(0, 1)$;
            **if** $p' < p$ **then**
                $c_n \sim \mathcal{C}$;
            **end**
        **end**
    **end**
    *// perform classifier update*
    $h \leftarrow \texttt{Fit}(h, X)$;
**end**

---

## C EXPERIMENTAL SETUP

We used PyTorch 1.5.0 (Paszke et al., 2019), OpenCV 4.2.0 (Bradski, 2000), and scikit-image 0.17.2 (van der Walt et al., 2014) for all experiments. In setting the number of epochs, we did not observe any significant degradation or improvements in performance when training for longer. We use fewer epochs in the case of Omniglot due to computational constraints, as the model is retrained for each test split.

For GTSRB, we use a 5-layer convolutional neural network adapted from the official PyTorch tutorials. To train with Picto, the data augmentation consists of PyTorch transforms RandomAffine(5, translate=(.15, .15), scale=(0.65, 1.05), shear=5), RandomPerspective(0.5, p=1); ColorJitter(brightness=.8, contrast=.8, saturation=.8, hue=.05); OpenCV box blur with a random kernel size between 1 and 6 in both dimensions (independently sampled, so not necessarily square); and a random exposure adjustment by adjusting all pixels by the same random amount between –30% and 50%. For refinement, we used step sizes of $\alpha = 2/255$ with 8 steps and an epsilon of $\epsilon = 4/255$ for the foreground only. For the observation function, we superimpose the segmented foreground of the transformed pictographic sign over the context. We train for 300 epochs using the Adam optimizer (learning rate 1e-4, weight decay 1e-4), with 5 examples of each class per batch and 20 batches per epoch. We report results for the model that achieves the best performance on the training set, checking every 5 epochs.

For MNIST, we use the two-layer convolutional neural network from the official PyTorch examples for MNIST, with Dropout regularization replaced with pre-activation BatchNorm. To train with Digit, the data augmentation consists of PyTorch transforms RandomAffine(15, translate=(.15, .15), scale=(0.75, 1.05), shear=40), RandomPerspective(0.5, p=1); OpenCV box blur with a random kernel size between 1 and 6 in both dimensions (independently sampled, so not necessarily square); then set the foreground to all pixels with value greater than 0.2. For refinement, we used step sizes of $\alpha = 1.6/255$ with 8 iterations and no projection ($\epsilon = \infty$). For the observation function, we blend the object with the context at a 2:1 ratio; this ensures that inputs have a well-defined ground truth label. We train for 300 epochs using the Adam optimizer (learning rate 1e-4, weight decay 1e-4), with 5 examples of each class per batch and 20 batches per epoch. We report results for the model that achieves the best performance on the training set, checking every 5 epochs.

For Omniglot, we use the pre-activation variant of ResNet18 (He et al., 2015). To train with Omnifont, we first preprocess with scikit-learn skeletonize and dilation to standardize stroke widths. Data augmentation consists of PyTorch transforms RandomAffine(15, translate=(.15, .15), scale=(0.75, 1.1), shear=20), RandomPerspective(0.25, p=1); OpenCV box blur with a random kernel size between 1 and 3 in both dimensions (independently sampled, so not necessarily square); then resize the images to 28 by 28. For refinement, we used step sizes of $\alpha = 1.6/255$ with 8 iterations and no projection ($\epsilon = \infty$). For the observation function, we blend the object with the context at a 2:1 ratio; this ensures that inputs have a well-defined ground truth label. For the $n$-way classification task, we randomly sample $n$ characters from the Omniglot test set, and use the corresponding characters from the Omnifont dataset as our training set. We then train a fresh model for 150 epochs using the Adam optimizer (learning rate 1e-4, weight decay 1e-4), and report performance on the all $20n$ images in the Omniglot test set, averaged over 20 runs (10 runs for the ablation studies).

# D  FULL EXPERIMENTAL RESULTS

We compare a model trained using our methods with previous state-of-the-art results from related settings using few-shot learning and domain adaptation on GTSRB (Table 2), MNIST (Table 3), and Omniglot (Table 4). When multiple experiments are reported for the same approach, we compare against both the most accurate result as well as the result using the least amount of target data. We distinguish between labelled (**L**) and unlabelled (**UL**) data; experiments for which the training data is not known are marked (**?**).

Table 2: GTSRB results.

| Approach | Method | Training Data | | Accuracy |
| --- | --- | --- | --- | --- |
| | | Source | Target | (%) |
| Baselines | Source Only (Saito et al. (2017)) | SynSign | | 79.2 |
| | Human (Stallkamp et al. (2012)) | | | 98.8 |
| | Target Only (Ganin et al. (2016)) | | All L | 99.8 |
| Few-Shot Learning | VPE (Kim et al. (2019))[§] | Picto[*] | 22 classes L | 83.8 |
| | MatchNet (Vinyals et al. (2016))[§] | | 22 classes L | 53.3 |
| | QuadNet (Kim et al. (2018))[§†] | | 22 classes L | 45.3 |
| Domain Adaptation | DSN (Bousmalis et al. (2016)) | SynSign | 1280 UL | 93.0 |
| | ML (Schoenauer-Sebag et al. (2019))[§] | SynSign | 22 classes L | 89.1 |
| | MADA (Pei et al. (2018))[§‡] | SynSign | 22 classes L | 84.8 |
| | DANN (Ganin et al. (2016)) | SynSign | 31367 UL | 88.7 |
| | ATT (Saito et al. (2017)) | SynSign | 31367 UL | **96.2** |
| Context Agnostic | baseline | Picto | | 72.0 |
| | + random-context | Picto | | 72.1 |
| | + refinement-only | Picto | | 86.4 |
| | + bias-correction | Picto | | 87.3 |
| | + full | Picto | | **95.9** |

[§]Test accuracy on remaining 21 unseen classes.

[*]Kim et al. (2019) use a pictographic dataset similar to Picto.

[†]Reported in Kim et al. (2019).

[‡]Reported in Schoenauer-Sebag et al. (2019).

Table 3: MNIST results.

| Approach | Method | Training Data | | Accuracy |
| | | Source | Target | (%) |
| --- | --- | --- | --- | --- |
| Baselines | Human (Netzer et al. (2011)) | | | 98.0 |
| | Target Only (Tzeng et al. (2017)) | | All L | 99.2 |
| Few-Shot Learning | FADA (Motiian et al. (2017)) | SVHN | 1 L / class | 72.8 |
| | + more data | SVHN | 7 L / class | 87.2 |
| | SiamNet (Koch (2015)) | Omniglot | 1 L / class | 70.3 |
| | MatchNet (Vinyals et al. (2016)) | Omniglot | 1 L / class | 72.0 |
| | APL (Ramalho and Garnelo (2019)) | Omniglot | 1 L / class | 61.0 |
| | + more data | Omniglot | ?[‡] | 86.0 |
| Domain Adaptation | DSN (Bousmalis et al. (2016)) | SVHN | 1000 UL | 82.7 |
| | DRCN (Ghifary et al. (2016)) | SVHN | ? | 81.9 |
| | DANN (Ganin et al. (2016)) | SVHN | ? | 73.9 |
| | ATT (Saito et al. (2017)) | SVHN | ? L + 1000 UL | 86.0 |
| | ADDA (Tzeng et al. (2017)) | SVHN | 60,000 UL | 76.0 |
| | CyCADA (Hoffman et al. (2017)) | SVHN | 60,000 UL | **90.4** |
| Context Agnostic | baseline | Digit | | 81.9 |
| | + random-context | Digit | | 88.3 |
| | + refinement-only | Digit | | 89.7 |
| | + bias-correction | Digit | | 89.2 |
| | + full | Digit | | **90.2** |

[‡]Cumulative accuracy from adapting over the test set.

Table 4: Omniglot results for one-shot classification.[‡]

| Approach | Method | Training Data | Accuracy (%) | |
| | | | 5-way | 20-way |
| --- | --- | --- | --- | --- |
| Baselines | Human (Lake et al. (2015)) | | | 95.5 |
| Few-Shot Learning | MANN (Santoro et al. (2016)) | Omniglot | 82.2 | |
| | SiamNet (Koch (2015)) | Omniglot | 96.7[§] | 92.0 |
| | MatchNet (Vinyals et al. (2016)) | Omniglot | 98.1 | 93.8 |
| | PN (Snell et al. (2017)) | Omniglot | 98.8 | 96.0 |
| | BPL (Lake et al. (2015)) | Omniglot | | 96.7 |
| | APL (Ramalho and Garnelo (2019)) | Omniglot | 97.9 | 97.2 |
| | RN (Sung et al. (2018)) | Omniglot | 99.6 | 97.6 |
| | MAML++ (Antoniou et al. (2018)) | Omniglot | 99.5 | 97.7 |
| | TapNet (Yoon et al. (2019)) | Omniglot | | 98.1 |
| | GCR (Li et al. (2019)) | Omniglot | **99.7** | **99.6** |
| Context Agnostic | baseline | Omnifont | | 71.9 |
| | + random-context | Omnifont | | 69.8 |
| | + refinement-only | Omnifont | | 90.8 |
| | + bias-correction | Omnifont | | 80.5 |
| | + full | Omnifont | **95.8** | 92.2 |

[‡]The exact set up of the one-shot classification task often varies between authors. We believe the broad performance numbers are still useful for contextualizing our approach, and refer the reader to the original works for details.

[§]As reported in Vinyals et al. (2016)

## E    TRAINING AND TEST SET VISUALIZATIONS

### E.1    DATASETS

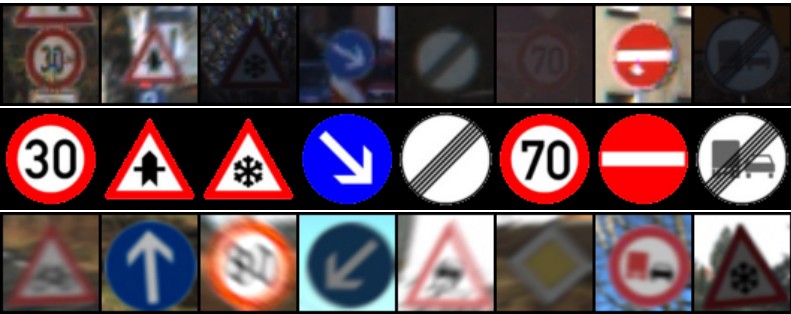

Figure 4: From top to bottom: samples from the GTSRB test set, Picto dataset, and SynSign dataset.

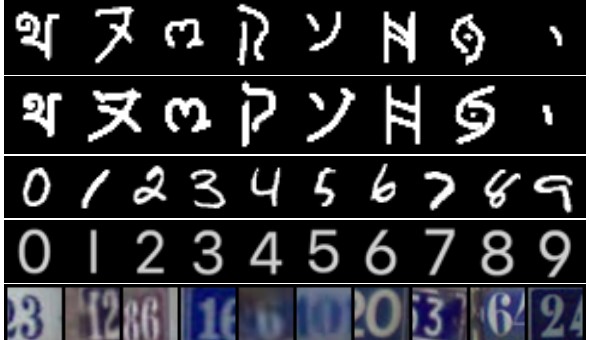

Figure 5: From top to bottom: samples from the Omniglot test set, Omnifot dataset, MNIST test set, Digit dataset, and SVHN training set.

### E.2 ABLATION STUDIES

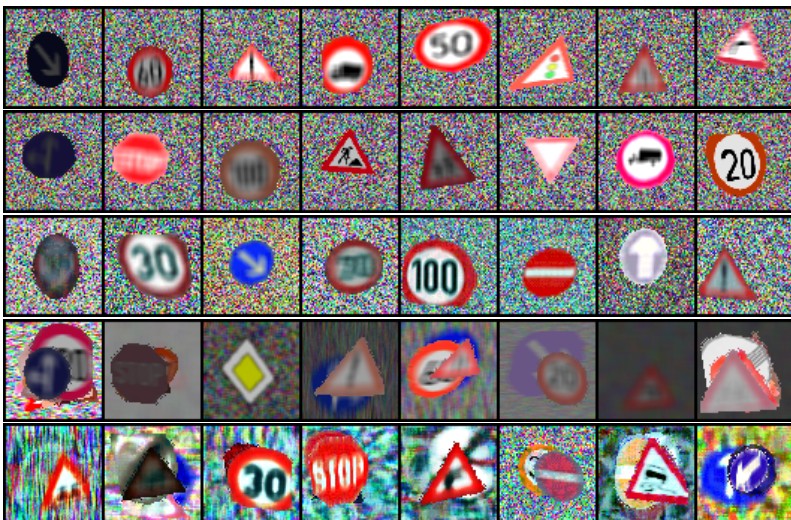

Figure 6: Training images from the first ablation study using Picto dataset. From top to bottom: baseline, random-context, refinement-only, bias-correction, full.

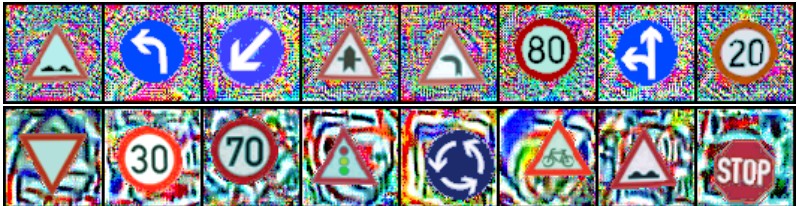

Figure 7: Test images from the second ablation study using the Picto dataset. Examples of test images generated using a PGD adversary initialized randomly (top) and with the bias heuristic (bottom) at $\epsilon = 255/255$.

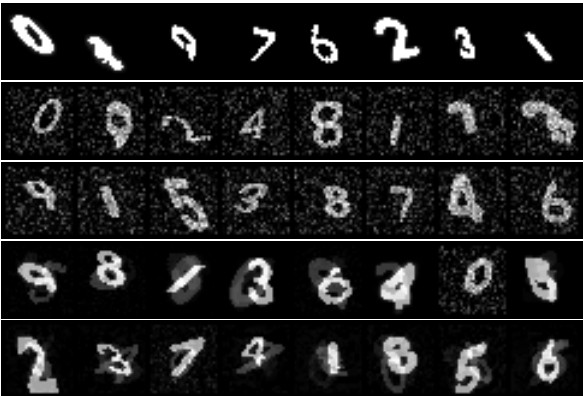

Figure 8: Training images from the first ablation study for the Digit dataset. From top to bottom: baseline, random-context, refinement-only, bias-correction, full.

