# OpenReview forum: "Context-Agnostic Learning Using Synthetic Data"
_ICLR.cc/2021/Conference — Reject_

### Official Review · AnonReviewer3 · 2020-10-27
**The paper proposes an interesting setting for learning, which relies on the ability to sample from the object and context spaces independently. This paper introduces a novel idea showing promising results in several benchmarks. The authors give a theoretical analysis for their method which is convincing. Extensive empirical results show that the method produces good results both in domain adaptation and few-shot learning settings.**

**Rating:** 6
**Confidence:** 4

**Review:**

Strengths:

1. This paper studies a theoretical setting for learning models whose predictions are independent of background signals. There are indeed some practical applications of this setting.

2. The proposed approach is very simple, yet effective. This method is able to learn a context-agnostic model by minimizing a formally defined notion of context bias.

3. On the theorical side, the authors try to explain their goal of classification as learning to extract reliable signals.

4. The paper is well written and easy to read. Also, the figures in supplementary materials help understand the paper, especially the context learning part.

Weaknesses/concerns:

1. The proposed method has a strict restriction: The training sets must have a single synthetic image for each object class with no additional information about the target domain. This may limit the training of the model in some hard conditions.

2. The quality of the paper will be upgraded if the authors further investigate more data augmentation or style transfer methods in related work and ablation study. Since the context and object concepts are similar with the style and content in some GANs and transferring works.

3. The experiments seem a little bit weak. The experiments are only conducted on two simple datasets, i.e. GTSRB and MINST.

4. The comparison with existing methods is insufficient. There is no representative baseline comparison. Though this setting is novel, we can conduct task by slighting changing existing methods.

---

> ### Author Response · Authors · 2020-11-15
> **1/1**
>
> We thank the reviewer for the insightful questions and comments. Please let us know if there are any points that require further elaboration.
>
> >  The proposed method has a strict restriction: The training sets must have a single synthetic image for each object class with no additional information about the target domain. This may limit the training of the model in some hard conditions.
>
> In fact, the only requirements for applying greedy bias correction are 1) a decomposition of the input space into object and context spaces, and 2) the ability to sample from the object and context spaces independently. In our heuristic, we suggest drawing samples from the object space uniformly at random; thus, it is straightforward to apply our techniques when there are multiple images (not necessarily synthetic) for each object class. Likewise, there is no restriction on using information from the target domain, though practically speaking, we argue that the ability of our approach to generalize even in absence of any target information is a significant strength of our approach.
>
> > The quality of the paper will be upgraded if the authors further investigate more data augmentation or style transfer methods in related work and ablation study. Since the context and object concepts are similar with the style and content in some GANs and transferring works
>
> Thank you for the suggestion! Style transfer does seem conceptually related, though GANs still require a large amount of images to learn, so we don't think they can be applied in the setting of our experiments. Note that we already cite several GAN-based approaches for domain adaptation--if there are any specific suggestions we would be happy to include them in the discussion.
>
> > The experiments seem a little bit weak. The experiments are only conducted on two simple datasets, i.e. GTSRB and MINST.
>
> Our results actually include a third dataset, Omniglot, which is a standard dataset for few-shot learning. This task differs from MNIST in that there are 1623 character classes, each of which is drawn by the same 20 people; the general idea is that Omniglot is about learning many characters over a small set of handwritings, whereas MNIST is about learning a small set of characters over many different handwritings.
>
> The datasets used for our experiments were selected because they very cleanly satisfy the theoretical assumptions of context-agnostic learning. In particular, we needed to be able to (1) produce uniform samples from the entire object space for training, and (2) have ground-truth segmentation of the object from the context for testing. These characteristics allow us to provide the results and ablation studies showing that, among other results:
> - classifiers trained using our method on synthetic datasets achieve both context-agnostic performance as well as generalization to the corresponding real datasets (Table 1)
> - classifiers trained using vanilla SGD on realistic datasets do not achieve context-agnostic performance (Figure 3)
>
> We would to emphasize that this second point is the second main contribution of this work, independent of the proposed Greedy Bias Correction. Specifically, we formalize the objective for context agnostic learning (and prove Theorem 3.1 as motivation for our setting). We believe this view on risk delivers a theoretically rigorous framework for analyzing the extent to which classifiers depend on background signals. Indeed, our ablation results show that image classifiers trained using vanilla SGD on real-world images are very brittle to small changes in the background. We note that the images generated when measuring the context agnostic performance are, qualitatively, extremely reasonable (see Figure 7 in Appendix E.2). This is not a trivial result, and we believe our datasets are a good setting for our theoretical contributions.
>
> > The comparison with existing methods is insufficient. There is no representative baseline comparison. Though this setting is novel, we can conduct task by slighting changing existing methods.
>
> We believe representative baseline comparisons are provided in our ablation studies. Our results include 1) standard data augmentation on our datasets, 2) the bias heuristic without local refinement, 3) local refinement without the bias heuristic.
>
> Appendix D, Full Experimental Results, also reports results from existing methods on the same test sets. They use varying types and amounts of additional data for training; we were unable to identify any existing methods that be applied to our training set. To be explicit, for GTSRB we train using 43 images in total (whereas the standard training set contains 39,209 images). Perhaps the reviewer could elaborate slightly on how to adapt existing methods to this setting?

---

### Official Review · AnonReviewer2 · 2020-10-28
**More comparison in experiments and motivation for this theoretical setting**

**Rating:** 5
**Confidence:** 3

**Review:**

The authors propose a theoretical setting for context-agnostic few-shot learning. While I do acknowledge that the background signals should not be involved or impact the predictions, I am concerned why this should have a new setting specific for this case? Can the authors elaborate more on the motivation for this theoretical setting and how this setting can benefit the application in the real world? To me, I appreciate that the synthetic data can provide more supervision for few-shot image classification tasks, while I don't understand why this should set up a new setting here.

In the experiments, the authors show the comparison to previous methods in Appendix Table 2, 3 , 4, and the proposed Context Agnostic method seems not outperforms previous baselines and sometimes even perform much worse than baselines. For example, 7% lower than GCR on Omniglot. Can the authors explain the  benefits of the proposed model please? Also, I notice the authors mainly use two simple types datasets, traffic signs and hand-written characters, while I am expecting to see the performance on more realistic datasets, such as ImageNet. Can the authors explain why they choose these two types of datasets? Are they easier to separate the context / background ?

As this paper seems more related to synthetic data, I am curious how synthetic data-based few-shot learning methods perform, such as [a,b]

[a]Eli Schwartz, Leonid Karlinsky, Joseph Shtok, Sivan Harary, Mattias Marder, Abhishek Kumar,
Rogerio Feris, Raja Giryes, and Alex Bronstein. Delta-encoder: an effective sample synthesis
method for few-shot object recognition. In S. Bengio, H. Wallach, H. Larochelle, K. Grauman,
N. Cesa-Bianchi, and R. Garnett (eds.), Advances in Neural Information Processing Systems 31,
pp. 2845–2855. 2018.

[b]Jian Zhang, Chenglong Zhao, Bingbing Ni, Minghao Xu, and Xiaokang Yang. Variational few-shot
learning. In Proceedings of the IEEE/CVF International Conference on Computer Vision (ICCV),
October 2019.

After rebuttal:
Thanks very much for the detailed response! I do agree with the AC's comment that *I don't see a drawback the use of only one sample and no info about the target, rather I'd like to know if the proposed method could use more than one sample in order to make the comparison with SoA methods fair and still be competitive or outperforming them.* Also,  I would recommend the authors to have more experimental validation and resubmission. Thus, I keep my score.

---

> ### Author Response · Authors · 2020-11-15
> **1/2**
>
> We thank the reviewer for the detailed questions and suggestions. If anything remains unclear, we would be happy to continue the discussion.
>
> > The authors propose a theoretical setting for context-agnostic few-shot learning. While I do acknowledge that the background signals should not be involved or impact the predictions, I am concerned why this should have a new setting specific for this case?
>
> The notion of a "background signal", and the question of whether the background is "involved or impact the predictions", are not well-defined. We develop a new setting so that we may speak precisely on these matters.
>
> > Can the authors elaborate more on the motivation for this theoretical setting...
>
> Our theoretical setting is one way to rigorously define the notion of a "background" signal by splitting the input space into separate object and context spaces. Theorem 3.1 justifies our chosen formalisms by drawing a link between our setting and the standard notion of risk using the context bias. This permits us to reason about ways to achieve this learning objective, and design experiments to evaluate algorithms for this objective.
>
> > ... and how this setting can benefit the application in the real world?
>
> For instance, the ablation studies show that classifiers trained using vanilla SGD on realistic datasets are surprisingly brittle to background signals (Table 3). This evaluation is well-defined, because we can measure against the objective of context-agnostic learning. We note that the images generated when measuring the context agnostic performance are, qualitatively, extremely reasonable (see Figure 7 in Appendix E.2). This is not a trivial observation, and we believe this result has significance for any computer vision systems that is deployed in the real world.
>
> > To me, I appreciate that the synthetic data can provide more supervision for few-shot image classification tasks, while I don't understand why this should set up a new setting here.
>
> First, our proposed approach (greedy bias correction) is an independent contribution from the theoretical setting (and Theorem 3.1), which defines the learning objective independently of any solution. Second, though our experiments use synthetic data (and also can be seen as a type of extremely limited few-shot learning), these are not inherent aspects of the algorithm.
>
> > In the experiments, the authors show the comparison to previous methods in Appendix Table 2, 3 , 4, and the proposed Context Agnostic method seems not outperforms previous baselines and sometimes even perform much worse than baselines. For example, 7% lower than GCR on Omniglot. Can the authors explain the benefits of the proposed model please?
>
> There are several benefits to our approach. The first is that we use significantly less data: only one sample of each class. For instance, our dataset for GTSRB consists of 43 images in total (whereas the standard training set contains 39,209 images). For Omniglot, we train using only the 20 (or 5) images in the test set. To the best of our knowledge, no other approach can be applied to this setting: both domain adaptation and few-shot learning (including GAN-based approaches) still require large amounts of training data. Thus, while a handful of other approaches achieve better test performance, our results are not directly comparable as the baseline approaches do so using significantly more data.
>
> Second, we note in the ablation studies that our models exhibit much higher resilience to background signals than models trained using standard supervised methods (Figure 3). As the existing baselines for few-shot learning and domain adaptation are not trained for the context agnostic objective, we believe they would suffer similarly.
>
> > Also, I notice the authors mainly use two simple types datasets, traffic signs and hand-written characters, while I am expecting to see the performance on more realistic datasets, such as ImageNet. Can the authors explain why they choose these two types of datasets?
>
> We selected datasets for which we could produce "ground truth" samples from the object space. This allows us a clean setting for both training our models, as well as measuring the context-agnostic performance. Note that none of the experiments in the paper use foregrounds separated from pre-existing training data; rather, we start with a single synthetic example of each class (e.g., for MNIST, we use a digit font; for GTSRB, we use the official, high-resolution images of each sign).
>
> > Are they easier to separate the context / background ?
>
> As far as we know, automatic segmentation is an open problem and current approaches still suffer from inaccuracies and artifacts. Thus we chose to avoid this approach for our experiments.

---

> > ### Author Response · Authors · 2020-11-15
> > **2/2**
> >
> > > As this paper seems more related to synthetic data, I am curious how synthetic data-based few-shot learning methods perform, such as [a,b]
> >
> > Thank you for the additional references, which we will be sure to include. [a] does not test on any of our benchmarks, but we note that their method requires large amounts of data for training the GAN. [b] benchmarks using the 20-way 1-shot task for Omniglot and achieves 98% accuracy vs 92% using our approach. However, they use 24000 images to learn the prior distribution, while we train our models from scratch using only a single example of each character in the test set.
> >
> > [a]Eli Schwartz, Leonid Karlinsky, Joseph Shtok, Sivan Harary, Mattias Marder, Abhishek Kumar, Rogerio Feris, Raja Giryes, and Alex Bronstein. Delta-encoder: an effective sample synthesis method for few-shot object recognition. In S. Bengio, H. Wallach, H. Larochelle, K. Grauman, N. Cesa-Bianchi, and R. Garnett (eds.), Advances in Neural Information Processing Systems 31, pp. 2845–2855. 2018.
> >
> > [b]Jian Zhang, Chenglong Zhao, Bingbing Ni, Minghao Xu, and Xiaokang Yang. Variational few-shot learning. In Proceedings of the IEEE/CVF International Conference on Computer Vision (ICCV), October 2019.

---

### Official Review · AnonReviewer4 · 2020-10-28
**Interesting paper**

**Rating:** 5
**Confidence:** 3

**Review:**

The paper defines the task of context-agnostic learning and proposes an algorithm to solve the problem while assuming the ability to sample objects and contexts independently. They propose to decompose factors contributing to the risk into two, context bias and object error. Based on this interpretation, an algorithm is designed to 'greedily correct bias' while employing adversarial training (or robustness training) for 'local refinement'. The method achieves high accuracy on two synthetic visual tasks, digits and traffic sign classification, when a model is trained using one sample per class from the source domain and tested on an unseen target domain.

+) Theorem 3.1 provides a new view on risk. Risk is decomposed into two factors, context bias and object error. I think this gives new insight to consider the effect of context bias and object modeling on risk separately.

+) The experimental results are impressive because the model is trained using a very limited number of samples (one sample per class from the source domain) but showed high generalization performance on an unseen domain. The proposed method achieves promising performance on two synthetic classification tasks compared to other existing methods for few-shot learning and domain adaptation, which requires more labeled or unlabeled data during training.

-) Their underlying assumption for greedy bias correction is that a classifier learns a strong bias on recent training inputs when taken as contexts. However, if stochastic gradient descent is used for optimization, I think it is unlikely because the model changes continuously. Therefore, it is uncertain how effective this greedy selection strategy can sample contexts with large bias.

-) Relating to the above point, I also have a concern about the experimental validation. In all the experiments, gamma is defined as a function that takes object and context images and outputs their overlap. It is not guaranteed that the proposed heuristic sampling strategy generalizes to other gamma functions.

-) Also, all the experiments are performed for a relatively small number of classes (up to 50), and synthetic images are small iconic images with objects in the center. Although the method shows promising results under this specific setting, it is hard to conclude that the proposed heuristics will generalize other settings, such as when there are more classes, image resolutions are higher, and objects have a larger variation in their appearance. I think evaluation on additional datasets with different characteristics (such as CIFAR-100, Caltech-256, CUB-200) would be necessary.

-) The assumption that one can sample objects and contexts independently may restrict its application.

Though I found this paper proposes an interesting view on risk, I would recommend 'reject' due to concerns stated above.

---
Thanks for the detailed response. I can agree that the observation function gamma in the form of addition can model many noisy signals. However, the argument in the paper that the proposed method works for an arbitrary gamma still lacks experimental validation. So I would like to keep my recommendation. My other concerns have been addressed.

---

> ### Author Response · Authors · 2020-11-15
> **1/2**
>
> We thank the reviewer for the detailed questions. Please see our responses below, and we would appreciate follow up discussions if anything remains unclear.
>
> > Their underlying assumption for greedy bias correction is that a classifier learns a strong bias on recent training inputs when taken as contexts. However, if stochastic gradient descent is used for optimization, I think it is unlikely because the model changes continuously. Therefore, it is uncertain how effective this greedy selection strategy can sample contexts with large bias.
>
> The greedy bias correction heuristic aims to solve an optimization objective which is computational intractable in general, namely, identifying the context with the highest bias. We do not claim that our heuristic successfully identifies contexts with the "largest" bias, but we believe our experimental results support that our heuristic identifies contexts of "sufficiently large" bias. First, as noted we achieve good performance on real-world datasets, and the ablation studies show our heuristic gives much better generalization than randomly sampling from the context space. Second, in Figure 3 we validate the empirical context bias of various classifiers by fixing the foreground and varying the background. In particular, initialization using our bias heuristic gives a much stronger attack than a random initialization, even when the attack is unbounded, suggesting that the contexts returned by the bias heuristic are indeed closer to the optimum.
>
> Regarding the interaction of our heuristic with SGD, the contexts are continuously updated along with the model, so we're not sure why SGD should present a particular challenge (perhaps the reviewer can clarify this point?)
>
> > Relating to the above point, I also have a concern about the experimental validation. In all the experiments, gamma is defined as a function that takes object and context images and outputs their overlap. It is not guaranteed that the proposed heuristic sampling strategy generalizes to other gamma functions.
>
> The reviewer raises a good point about the observation function being similar across experiments. We do not think this is a significant weakness however, as many types of noise manifest additively in the signal, particularly noise which arises from natural observations (e.g., the background of images, noise in audio clips, measurement noise for sensors, etc.). For instance, the Kalman filter is another example of a model that assumes additive noise, but this has not prevented it from being widely used (perhaps most famously in the navigation systems of manned missions to the moon).
>
> > Also, all the experiments are performed for a relatively small number of classes (up to 50), and synthetic images are small iconic images with objects in the center. Although the method shows promising results under this specific setting, it is hard to conclude that the proposed heuristics will generalize other settings, such as when there are more classes, image resolutions are higher, and objects have a larger variation in their appearance. I think evaluation on additional datasets with different characteristics (such as CIFAR-100, Caltech-256, CUB-200) would be necessary.
>
> The datasets used for our experiments were selected because they very cleanly satisfy the theoretical assumptions of context-agnostic learning. In particular, we needed to be able to (1) produce uniform samples from the entire object space for training, and (2) have ground-truth segmentation of the object from the context for testing. These characteristics allow us to provide the results and ablation studies showing that, among other results:
> - classifiers trained using our method on synthetic datasets achieve both context-agnostic performance as well as generalization to the corresponding real datasets (Table 1)
> - classifiers trained using vanilla SGD on realistic datasets do not achieve context-agnostic performance (Figure 3)
>
> We would to emphasize that this second point is the second main contribution of this work, independent of the proposed Greedy Bias Correction. Specifically, we formalize the objective for context agnostic learning (and prove Theorem 3.1 as motivation for our setting). We believe this view on risk delivers a theoretically rigorous framework for analyzing the extent to which classifiers depend on background signals. Indeed, our ablation results show that image classifiers trained using vanilla SGD on real-world images are very brittle to small changes in the background. We note that the images generated when measuring the context agnostic performance are, qualitatively, extremely reasonable (see Figure 7 in Appendix E.2). This is not a trivial result, and we believe our datasets are a good setting for our theoretical contributions.

---

> > ### Author Response · Authors · 2020-11-15
> > **2/2**
> >
> > > The assumption that one can sample objects and contexts independently may restrict its application.
> >
> > Our technique for context-agnostic learning is immediately applicable to synthetic data pipelines, which is widely used in, e.g., autonomous driving and robotics (see [1] for a survey).
> >
> > [1] Sergey I. Nikolenko, Synthetic Data for Deep Learning. https://arxiv.org/abs/1909.11512

---

### Official Review · AnonReviewer1 · 2020-10-29
**How to achieve "context-agnostic" characteristic in algorithm 1 and 2?**

**Rating:** 7
**Confidence:** 3

**Review:**

This paper proposed a context-agnostic learning approach that combines an object area and a context image (s.t. background image) to generate input synthetic image and train the model as context-independent. The proposed method is made more efficient by including this generation process in the training loop, compared to a exhaustive sampling method that randomly selects from a set of combinations of object areas and context images. When applying this method on the task of traffic sign recognition, character recognition, it provides enhanced performance in a setting where model is trained using synthetic data and evaluated on real-world dataset.

My concern is the presentation.
"Context" and "context-agnostic" are properly defined. However, it is very difficult to understand how to achieve "context-agnostic" in algorithm 1 and 2. In my understanding, context bias is corrected by forcing a model trained using the opposite (or other) object class in the same context in the next training iteration. If it is correct, how can the bias be corrected just by training with other objects? Does it affect to reduce ||B(h,c)|| in Definition 3.3? Please clarify it.

In addition, the effectiveness of the proposed method can be verified with such simple tasks, but it is advisable to add a discussion about more complex tasks such as object detection that requires more diverse contexts. This will be more interesting for many.

I can recognize some technical contributions that can appeal to many ICLR researchers.

As a minor correction,
In Section 3.2, the object error o^ is defined but not used anywhere.

----------------------------------------------------------------------------------------------------------
I have read the revised manuscript and the comments of all reviewers. For concerns that the experiments are not sufficient to validate the proposed method, I am leaning to the author's rebuttals that the datasets have been chosen to meet the heretical assumptions of context-agnostic learning. The given datasets seem to be sufficient to demonstrate the effectiveness of the proposed learning method.

Therefore, I will not change the rating.

---

> ### Author Response · Authors · 2020-11-15
> **1/1**
>
> We thank the reviewer for the suggestions. Please let us know if the following responses leave any concerns unaddressed.
>
> > My concern is the presentation. "Context" and "context-agnostic" are properly defined. However, it is very difficult to understand how to achieve "context-agnostic" in algorithm 1 and 2. In my understanding, context bias is corrected by forcing a model trained using the opposite (or other) object class in the same context in the next training iteration. If it is correct, how can the bias be corrected just by training with other objects? Does it affect to reduce ||B(h,c)|| in Definition 3.3? Please clarify it.
>
> The rationale behind our approach is covered at the end of Section 3.1. The naive method for controlling the bias with respect to Theorem 3.1 would be: for each training sample use the context of the current maximum bias, i.e., argmax_c ||B(h,c)||. Unfortunately, as we note, this optimization problem is computationally intractable in general. Our heuristic is based on the observation that the most recent training sample can be repurposed as a context of large bias for the next training sample.
>
> We believe our experiments support this heuristic. First, we are able to generalize from synthetic to several real-world datasets, and our ablation studies show that this heuristic contributes significantly to achieving such performance (Figure 1). We also compare our classifiers against training directly on real-world data, and show that our classifiers are far more robust to background perturbations than the classifiers trained using real-world data, i.e., our method does indeed succeed in producing classifiers which are empirically context-agnostic (Figure 3). This evaluation further demonstrates that initialization using our bias heuristic gives a much stronger attack than a random initialization, even when the attack is unbounded, suggesting that the contexts returned by the bias heuristic are indeed closer to the optimum.
>
> > In addition, the effectiveness of the proposed method can be verified with such simple tasks, but it is advisable to add a discussion about more complex tasks such as object detection that requires more diverse contexts. This will be more interesting for many.
>
> Object detection is an interesting extension. For now, we have focused on classification because it makes the theoretical development cleaner, and we felt it was important that both the setting and results be easily interpretable. However the suggestion to include a section discussing more complex tasks is well-received and we will use a portion of the additional page limit to this.
>
> > As a minor correction, In Section 3.2, the object error o^ is defined but not used anywhere.
>
> The object error is used in the statement of Theorem 3.1--we will address this in our revision.

---

### Comment · Area_Chair1 · 2020-11-20
**Please, check rebuttals and start discussion if needed**

Dear Reviewers and Authors,
Thanks for starting the discussion.

Reviewers: please, check the rebuttals provided by the authors, verify if they replied properly and you are satisfied.
Possibly, give further feedback or make questions, only if needed and important for your final evaluation.
Please, be accurate and precise in your further requests, so that authors can understand and reply properly and focused.
Eventually, you need to revise your review and report final comments and rating.

Authors: please, check if there are further clarifications needed by the Reviewers.
Please, be focused in your final answers and avoid to ask questions to Reviewers, if not absolutely necessary.

For All: please, I would avoid a long chat-like discussion, a couple of iterations are affordable on a few specific points to be clarified, but no more.

Thanks and best regards

AC

---

### Decision · Program_Chairs · 2021-01-07
**Final Decision**

**Decision:**

Reject

**Comment:**

The paper addresses the task of context-agnostic learning and presents an algorithm to solve the problem while assuming the ability to sample objects and contexts independently. It is reported a theoretical ground proposing to decompose factors contributing to the classification risk in context bias and object error. The method makes use of only one synthetic sample in training, still being able to generalize well.

The paper received contrasting reviews, 2 positive (7 and 6) and 2 below threshold (5 and 5). R2, R3 and R4 raised similar issues, especially regarding the experimental validation, which is the main shortcoming of the work: addressing "simple" datasets only, no comprehensive comparative analysis only in relation to baselines (vanilla SGD) but not in relation to state-of-the-art methods, possibly slightly revised to accomplish the experimental protocol proposed in this work (authors claim that the originality of the work do not allow a proper comparison with SoA method as-is). Indeed, I deem R3's rating (6) a bit overestimated given the provided comments.
AC does not see an issue the use of only one sample and no info about target, rather, it'd be interesting to know if the proposed method could use more than one sample in order to make the comparison with SoA methods fair, while assessing performance in comparative terms with SoA, to give value to the method also in relation to performance.

Unfortunately the rebuttal did not lead an increase of the ratings, nor to better comments.
After the rebuttal, R2 and R4 still remained below threshold; R1 was also not changing idea, remaining positive, and R3 did not react after rebuttal.

Overall, the AC deems this paper containing interesting contributions, but it is not sufficiently ready to be accepted at ICLR mainly because the experiental validation is not showing a fully convincing evaluation of the proposed approach (see above).